# GRAPH GENERATION VIA TEMPORAL-AWARE BIASED WALKS

## ABSTRACT

Some real networks keep a fixed structure (e.g., roads, sensors and their connections) while node or edge signals evolve over time. Existing graph generators either model topology changes (i.e., edge additions/deletions) or focus only on static graph properties (such as degree distributions or motifs), without considering how temporal signals shape the generated structure. By approaching the problem from an unconventional perspective, we introduce temporally attributed graphs, named TANGEM, that integrate a temporal similarity matrix into biased random walks, thereby coupling signals with structure to generate graphs that highlight patterns reflecting how nodes co-activate over time. We evaluate TANGEM using an approach that separates structural fidelity (clustering, spectral metrics) from downstream temporal consistency, allowing us to clearly isolate the impact of the topology generator itself. In time series benchmarks, TANGEM consistently outperforms strong baselines in structural metrics while remaining lightweight, learning from a single graph. These results show that adding temporal bias to structural sampling produces more realistic graphs and establishes TANGEM as a basis for future models that further integrate evolving signals and structure.

## 1 INTRODUCTION

Graphs provide a powerful abstraction for representing relational information across diverse domains, including social networks (Leskovec & Mcauley, 2012), biological and molecular structures (Stark et al., 2005), recommender systems, and infrastructure networks such as roads, power grids, and computer systems (Li et al., 2018; Olug et al., 2024). Generative models of graphs, which learn underlying graph distributions from data, are also increasingly important for protein design, drug discovery (Liu et al., 2019), and various networked systems, such as the Internet of Things (De et al., 2022), as well as data augmentation approaches (Bas et al., 2024).

In many of these settings, networks are not only structured but also timeful: nodes and edges may appear or disappear, and their attributes often fluctuate (Rozemberczki et al., 2021).While several recent graph generation studies (e.g., TagGen (Zhou et al., 2020), TIGGER (Gupta et al., 2022), DAMNETS (Clarkson et al., 2022)) tackle the case where the graph topology itself evolves, which is an important and timely contribution, far less attention has been given to the complementary, and practically critical regime where the network topology remains fixed but the node or edge signals change substantially over time. For example, the road infrastructure of a city is largely fixed, yet traffic intensities evolve dynamically; citation networks maintain a stable core of authors and affiliations, but citation frequencies shift over time; sensor networks retain their physical layout, yet the measurements vary continuously; these are only a few such cases.

We argue that even with a fixed topology, temporal dynamics should shape the generated connectivity patterns (e.g., which motifs recur or which spectral modes are emphasized) because time shapes how nodes co-activate. However, existing approaches typically fall into three separate tracks: methods that emphasize topological evolution through edge additions and deletions (Zhou et al., 2020; Gupta et al., 2022; Clarkson et al., 2022); generation models that focus solely on structural graph patterns without temporal context (Bojchevski et al., 2018; Simonovsky & Komodakis, 2018; You et al., 2018b; Shi et al., 2020); and forecasting approaches that predict node signals on a fixed graph but do not allow temporal dynamics to inform the structural patterns of the generated graphs (Islam et al., 2024; Yu et al., 2018; Li et al., 2018).

To address this gap, we propose TANGEM (Temporally Attributed Network GEneration Model), inspired by "tandem", *two components working together*. TANGEM bridges *signal* and *structure* through a temporal-aware biased random walk. We learn a pairwise similarity matrix $\rho$ from historical node signals and inject it as a time-aware bias into a second-order walk, which is modeled autoregressively with a transformer. This design allows temporal co-activation to steer which motifs and spectral modes are emphasized during graph generation, an aspect overlooked by the current literature. To our knowledge, TANGEM is the first generator of temporally attributed graphs with a fixed topology that explicitly couples temporal signals with walk sampling. Our main contributions and findings are: i) a learned similarity matrix $\rho$ computed from historical signals and injected as a time-aware bias into an autoregressive, transformer-based random walk, explicitly coupling temporal co-activation with structural sampling; (ii) empirical gains on temporally attributed benchmarks in structural fidelity, particularly clustering coefficients and spectral distributions, showing that temporal bias in structural sampling better captures recurring interaction patterns; and (iii) ablation studies and clear evaluation, separating structural fidelity (e.g., clustering and spectral MMD) from downstream temporal consistency [1].

## 2 RELATED WORK

**Deep graph generators.** Modern deep graph generators mitigate limits in the probabilistic graph generators (e.g., Erdős–Rényi, Barabási–Albert, (Erdös & Rényi, 1959; Albert & Barabási, 2002) and stochastic block models (Holland et al., 1983; Mandjes et al., 2019)) by learning from examples. These models can be broadly categorized into three frameworks: generative adversarial networks (GANs) (Bojchevski et al., 2018), variational auto-encoders (VAEs) (Simonovsky & Komodakis, 2018), and autoregressive models (You et al., 2018b). There are also other methods that leverage reinforcement learning (You et al., 2018a) and flow-based learning (Shi et al., 2020) approaches, mainly for molecular graph generation. Concurrently, there is rapid progress on diffusion and score-based graph generators. Score models tackle the isomorphism issue by operating on set-valued states or exchangeable representations (Niu et al., 2020; Jo et al., 2022). Diffusion-style methods (discrete denoising or continuous-time variants) such as DiGress (Vignac et al., 2023), discrete-state continuous-time diffusion (Xu et al., 2024), CoMeTh (Xu et al., 2024), and DeFoG (QIN et al., 2025) achieve high-fidelity topology and attribute generation, especially in molecular settings (i.e., relatively small graphs), but they still optimize static fidelity (and often require large graph corpora and multi-step sampling). These approaches, however, focus almost exclusively on topology, optimizing adjacency-level fidelity (or delta matrices (Clarkson et al., 2022)) and typically ignore temporal node/edge signals that co-evolve with structure.

**Temporal graph learning.** A parallel line of work deals with temporal graph learning rather than graph generation, focusing on predicting signals that evolve on a fixed or slowly changing network. Models such as T-GCN (Zhao et al., 2019) and ST-GCN/DCRNN (Yu et al., 2018; Li et al., 2018) combine graph convolution with recurrent or diffusion mechanisms to forecast traffic and other time-series data at network nodes. More recent efforts such as DyGCL (Islam et al., 2024) extend this idea to event prediction by learning dynamic graph representations with contrastive objectives. Although these approaches show how temporal information can improve forecasting on graphs, they are not designed to generate graphs. Nevertheless, they reveal two well-established directions in graph learning that highlight the bidirectional dependency between structure and signals: (1) structure to signals, where topology aids forecasting (Yu et al., 2018; Li et al., 2018); and (2) signals to structure, where temporal co-variation informs graph inference (Dong et al., 2016; Thanou et al., 2016). Together, these studies underscore the need for models that jointly capture structure and temporal dynamics, serving as a direct motivation for the approach we develop in this paper.

**Dynamic graph generation with evolving topologies.** Another parallel line of work focuses on generating dynamic graphs by modeling how the topology evolves over time. Temporal interaction models like TagGen and TIGGER create timestamped edges using temporal random walks and discriminators or inductive embeddings, explicitly adding and deleting edges over time (Zhou et al., 2020; Gupta et al., 2022). DAMNETS predicts edge additions, removals, or persistence between

---

[1]For completeness, we *transfer corresponding node features* to the generated topology and then apply autoregressive updates (GCN+RNN), confirming that the observed structural gains arise from the proposed topology module rather than from the forecasting side. This routine component is *not* a technical contribution and is detailed in the Appendix A.9

snapshots using a learned delta matrix (Clarkson et al., 2022). Motif-oriented approaches such as DyMOND replicate evolving motif patterns but still treat edge activity as the main signal and generally ignore node or edge attributes (Zeno et al., 2021). These works mainly assume or prioritize structural changes and tend to overlook temporal node features as primary modeling targets.

In contrast, TANGEM (i) jointly models structure and signals rather than treating time as an add-on; (ii) differs from dynamic graph generators (e.g., TagGen, TIGGER, DAMNETS) by not modeling edge additions or deletions but instead focusing on signal–structure coupling within a fixed topology graph generation setting; and (iii) complements deep graph generators by providing a lightweight, history-conditioned mechanism, learned from a single graph.

## 3 METHODOLOGY

In this section, we first define a temporally attributed graph and then describe our method.

**Definition 1** (Temporally-Attributed Graph). *A temporally-attributed graph $\mathcal{G}$ is defined as $\mathcal{G} = (\mathbf{G}, \mathbf{X})$, where $\mathbf{G} = (V, E)$ represents the underlying graph topology with node set $V$ and edge set $E$, and $\mathbf{X} \in \mathbb{R}^{|V| \times T \times F}$ is the tensor of temporal node features, with $T$ denoting the number of discrete time steps and $F$ the number of features per node per timestamps. This formulation captures the evolution of node attributes over time, preserving the structural relationships encoded in $\mathbf{G}$.*

Prior research has demonstrated the effectiveness of biased random walks in capturing graph topology. Specifically, Bojchevski et al. (2018) highlighted that increasing the number and length of these walks improves the ability of the generative model to learn graph representations. However, scale-free networks, such as social networks and citation networks, pose unique challenges, as their number of edges grows faster than their number of nodes due to the preferential attachment mechanism Barabási et al. (2002); Lattanzi & Sivakumar (2009). Consequently, the number of walks required for effective representation increases superlinearly with graph size and even quadratically for certain types of scale-free networks Barabási et al. (2002), resulting in significant computational inefficiencies. Additionally, sampling an excessive number of uniform random walks raises the risk of the generative model overfitting by memorizing the local patterns of the input graph rather than generating meaningful synthetic variations. These challenges underscore the need for strategically designed walks with an ultimate focus on generation, rather than simply increasing the quantity of uniform random walks. To address this, we re-design the biased random walk proposed in Node2Vec Grover & Leskovec (2016) with emphasis on structural exploration and with an additional bias obtained via the temporal node attributes, which leverages the principle of homophily. The ultimate motivation is to capture as many connections as possible (i.e., low repetitions of nodes during walk) while not sacrificing the coherence of local structures and the meaningful community organization of the network.

**Temporal Aware Biased Random Walk.** To incorporate the temporal (i.e., dynamic) attributes of nodes into a graph's structure, we must make foundational assumptions about how these features are distributed throughout the graph. The principle of spatial smoothness assumes that the features of nodes that are connected and structurally close in a graph (e.g., being in the same community) are more alike than those of disconnected nodes Dong et al. (2016). When we consider features that evolve over time, we extend this assumption to state that the temporal correlation of features is stronger between these same structurally closer nodes. This means a node's feature trajectory is more likely to mirror that of its neighbors within its own community than that of its neighbors in other communities, as their shared environment and interactions lead to more similar and coordinated changes over time.

Under this assumption, we introduce a temporal bias into the random walk process. In the proposed temporal-aware biased random walk method, the walk is guided not only by structural connectivity (i.e., Node2Vec Grover & Leskovec (2016)) but also by a preference to move to nodes whose temporal features are similar to the current node's features. This encourages the random walk to stay within temporally coherent regions of the graph, thereby capturing both the graph topology and temporal dynamics simultaneously. We apply this by defining a pairwise similarity matrix $\mathbf{S} \in \mathbb{R}^{|V| \times |V|}$, where $s_{ij}$ measures the temporal alignment between vertices with indices $i$ and $j$. This matrix captures the interactions between temporal features from the temporal features of node matrix $\mathbf{X} \in \mathbb{R}^{|V| \times T \times F}$. The similarity is defined by a metric $f$ as in Eq. 1.

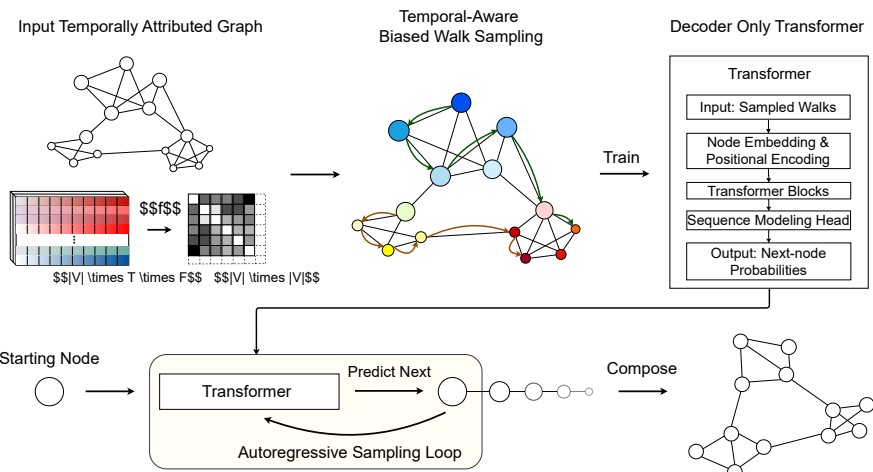

Figure 1: Overview of TANGEM. We first compute a $|V| \times |V|$ temporal correlation matrix from the node attributes, which captures how nodes relate to each other across time. This matrix, together with a structural bias that encourages exploratory behavior, guides the walk process used to sample node sequences. The sampled walks are then used as input to a decoder-only transformer, which is trained to model node sequences. During inference, we begin from an initial node (either chosen randomly or specified) and generate a walk autoregressively until the maximum sequence length is reached. The resulting node sequence is finally converted back into a graph structure, using consecutiveness information.

$$S_{ij} = f(x_i, x_j) \tag{1}$$

In previous work, node signals $\mathbf{X}$ are used to *infer* an unknown graph structure by estimating an adjacency matrix $\mathbf{A}$ or a Laplacian $\mathbf{L}$ through constrained optimization (e.g., (Kalofolias, 2016; Thanou et al., 2016)). In contrast, our setting assumes a known, fixed topology $\mathbf{G} = (V, E)$ and does *not* attempt to reconstruct $\mathbf{A}$ or $\mathbf{L}$. Instead, we compute a pairwise temporal similarity matrix $\mathbf{S} \in \mathbb{R}^{|V| \times |V|}$, where each entry $S_{ij} = f(x_i, x_j)$ measures the alignment between the temporal node features $x_i, x_j \in \mathbb{R}^{T \times F}$. This similarity matrix $\mathbf{S}$ directly encodes temporal interactions among vertices and is used as a bias in our random-walk–based graph generation framework, without requiring explicit topology reconstruction.

The traditional second-order random walk is characterized by two parameters, $p$ (return parameter) and $q$ (in-out parameter), to guide the walk. Given a walker just traversed an edge between node $k$ and node $u$, defining the $\pi_{uv}$ as unnormalized transition probability between node $u$ and node $v$, we get the Eq. 2 for unweighted graphs:

$$\pi_{uv} = \alpha_{pq}(k, v) \tag{2}$$

where the search bias $\alpha$ is defined as

$$\alpha_{pq}(k, v) = \begin{cases} \frac{1}{p} & \text{if } \delta_{kv} = 0 \\ 1 & \text{if } \delta_{kv} = 1 \\ \frac{1}{q} & \text{if } \delta_{kv} = 2 \end{cases} \tag{3}$$

The $\delta_{kv}$ in Eq. 3 is the shortest path between node $k$ and node $v$, and is defined as Eq. 4.

$$\delta_{kv} = \begin{cases} 0 & \text{if } k = v \\ 1 & \text{if } (k, v) \in E \\ 2 & \text{if } (k, v) \notin E \end{cases} \tag{4}$$

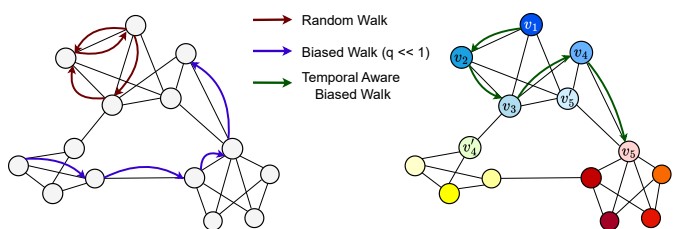

Figure 2: The illustration compares three types of walks. In a uniform random walk, the walker moves randomly at each step but often remains in the same neighborhood due to frequent backtracking. A biased walk with exploratory behavior ($q \ll 1$) instead prioritizes moving away from the current neighborhood, reducing redundancy and encouraging exploration of the graph structure. A temporal-aware biased walk balances these two behaviors by always moving for exploration while giving priority to nodes with strong temporal correlation (represented by similar colors).

According to our observations, encouraging the depth-first approximation with lower $q$ values improves the learning of graph topology, particularly for generation purposes. It is admissible because the lower values of $q$ reduce the frequency of edge repetitions during the walk, which is desired for the generative model to capture more information on connectivity structure of the graph. Considering this, we update the Eq. 3 by eliminating the parameter $p$ (fixing it as 1), and adding a temporal bias function $\rho(u, v)$ where the $u$ and $v$ are source and target nodes, respectively. With lower values of parameter $q$, this formulation benefits to sampling explorative walks, while still taking into account the locality information, thanks to the temporal bias function $\rho$. Finally, we obtain Eq. 5.

$$\alpha_q(k, v, u) = \begin{cases} \frac{1}{q}\rho(u, v) & \text{if } \delta_{kv} = 2 \\ 1 & \text{otherwise} \end{cases} \tag{5}$$

The $\rho(u, v)$ is simply a lookup function as Eq. 6.

$$\rho(u, v) = S_{uv} \tag{6}$$

Then, to consider the two-hop neighbors, $\rho(u, v)$ can be updated as Eq. 7.

$$\rho(u, v) = S_{uv} + \lambda \frac{\sum_{x \in \mathcal{N}(v) \setminus (\mathcal{N}(u) \cup \{u\})} S_{ux}}{|\mathcal{N}(v) \setminus \mathcal{N}(u)| - 1} \tag{7}$$

Here, $\mathcal{N}(.)$ is the set of neighbors of the given node, and the $\lambda$ is a hyperparameter to control the influence of second-hop neighbors (i.e., neighbors of $v$, that are not directly connected to $u$). The set $|\mathcal{N}(v) \setminus \mathcal{N}(u)|$ refers to the size of the set difference between the set of neighbors of vertex $v$ and the set of neighbors of the vertex $u$ in a graph.

For intuition, Fig. 2 illustrates how temporal features are incorporated into the walk sampling procedure. Suppose the walker is at $v_3$; the exploratory bias reduces the chance of visiting $v_1$ or $v_5'$ (Eq. 3), but does not distinguish between $v_4$ and $v_4'$; here, the temporal bias favors $v_4$, which lies in a temporally coherent region. Conversely, when the walker is at $v_4$, the exploratory bias discourages revisiting $v_5'$, which is already connected to the previous node $v_3$, and instead encourages moving forward to $v_5$, which promotes exploration.

**Autoregressive Generation via Sequence Modelling.** We treat the generation of the graph structure as a sequence generation task, where the sequence refers to a series of nodes. Compared to sampling the graph in a single step, sequential representation is often desirable, as it allows for autoregressive modeling of the graph, enabling the capture of complex joint probability of nodes and connections via recurrent learning architectures such as RNNs and transformer-based models You et al. (2018b); Bojchevski et al. (2018); Liao et al. (2019). In contrast, sampling the adjacency matrix (or an equivalent) in a single step suffers from scalability issues, as the adjacency matrix size grows quadratically with the number of vertices Tavakoli et al. (2017). For these two reasons, we

use a transformer model to learn the underlying marginal distribution of the joint distribution of the graph (the same form given in Eq. 8) by leveraging temporal-aware biased random walks sampled from the graph. In other words, we take into account all tokens generated so far, not just the current state, when computing the next-step probability.

At the inference time (i.e., generating synthetic graphs), we start with an initial token and generate a sequence of nodes conditioned on the previous tokens, following the approach used in sequence models, similar to language models. The joint probability of generating a sequence can be decomposed as in Eq. 8, where the $v_1$ is the starting token (i.e., first node), and $\mathbf{s}_{<m}$ is the sequence generated so far. Here, $P(\mathbf{s})$ denotes the overall probability of generating the full node sequence. The term $P(v_1)$ represents the probability of generating the initial node (or starting token). Each subsequent term $P(v_m \mid \mathbf{s}_{<m})$ corresponds to the probability of generating the $m$-th node $v_m$, conditioned on the previously generated sequence $\mathbf{s}_{<m} = \{v_1, \ldots, v_{m-1}\}$

$$P(\mathbf{s}) = P(v_1) \prod_{m=2}^{M} P(v_m \mid \mathbf{s}_{<m}) \tag{8}$$

We use the terms *sequence* and *walk* interchangeably, with the hyperparameter $M$ denoting both the sequence length and number of generated tokens. Unlike prior approaches such as (Bojchevski et al., 2018) and (Zhou et al., 2020), which rely on aggregating multiple short walks to form an adjacency matrix, we instead generate a single or a few long walks. This design choice is motivated by the ability of the transformer model to capture long-range dependencies within sequences. The resulting sequence directly encodes graph connectivity, uniquely determining the graph structure. Notably, if one sequence of length $M$ is used, the generated graph contains at most $M$ unique nodes, and exactly $M$ only for a path graph, since repeated vertices in the sequence reduce the effective node count.

## 4 EXPERIMENTS

We evaluate TANGEM on synthetic and real-world temporal datasets, benchmarking structural fidelity (degree, clustering, spectral, motif, orbit) against strong baselines while isolating the effect of our topology generator, and we also compare four walk-sampling strategies.

### 4.1 DATASETS

We evaluate TANGEM on both synthetic and real-world temporal datasets with varying sizes and properties (e.g., grid-like, path-like, community).

**IBB dataset.** The IBB graph datasets consist of road traffic data collected in Istanbul (Olug et al., 2024). We use two real-world temporal graph datasets with sizes $|N| = 256$ and $|N| = 294$. These graphs differ in both structural characteristics and overall size. In each graph, vertices represent specific locations, and the associated temporal attributes correspond to traffic density values recorded at those locations. The two datasets, IBB1 and IBB2, correspond to different regions of Istanbul, forming distinct subgraphs.

**PEMS dataset.** The PEMS-04 traffic dataset (Chen et al., 2001) was collected in a major metropolitan area in California. The original dynamic graph has a size of $|N| = 307$. For our experiments, we focus on the largest connected component, which consists of $|N| = 237$ nodes.

**CiteSeer dataset.** The CiteSeer (CS) network (Sen et al., 2008) is a widely used benchmark dataset where documents are represented as nodes and citation relationships as edges. Following (Bojchevski et al., 2018), we used the largest connected component, which has a size of $|N| = 2120$. To generate synthetic temporal node features and introduce dynamicity, we used a heat diffusion process (see the Appendix A.6 for details).

### 4.2 BASELINE MODELS

We benchmarked TANGEM against both classical and modern graph generators. As traditional baselines we used the Erdős–Rényi (E–R) model (Erdos & Renyi, 1959), which forms edges with

Table 1: Comparison of TANGEM to other generative models across different datasets and metrics (lower values are better). TANGEM-Plain is trained on uniform random walks, while TANGEM is trained on biased and temporal-aware walks formulated in Eq. 5. Bold numbers correspond to the best performance, while underlined entries are the second-best performance.

| | IBB1 | | | | | IBB2 | | | | |
|---|---|---|---|---|---|---|---|---|---|---|
| | Degree | Clustering | Spectral | Orbit | Motif | Degree | Clustering | Spectral | Orbit | Motif |
| E-R Generator | 0.1464 | 0.0508 | 0.0862 | 0.1887 | 0.9817 | 0.1809 | 0.3898 | 0.0640 | 0.7971 | 1.2489 |
| B-A Generator | 0.4117 | 0.0001 | 0.2769 | 1.2684 | 1.5436 | 0.3033 | 1.4708 | 0.1262 | 1.2303 | 1.3013 |
| GraphVAE | 0.2952 | 1.2321 | 0.1592 | 1.3620 | 1.2369 | 0.3138 | 1.2748 | 0.1085 | 1.3574 | 1.2164 |
| NetGAN | 0.1316 | 0.0780 | 0.1037 | 0.1693 | 1.0169 | 0.2803 | 0.8705 | 0.0800 | 1.4898 | 1.7207 |
| GraphRNN | 0.0165 | 0.0527 | 0.0988 | 0.0057 | 0.2143 | 0.2243 | 0.1295 | 0.1266 | 0.3793 | 1.5659 |
| GRAN | 0.0153 | 0.0412 | 0.1180 | 0.0160 | 0.5055 | **0.0358** | 0.5573 | 0.0509 | 0.0259 | 0.5599 |
| DiGress | 0.0355 | 0.0071 | 0.1108 | 0.0073 | 0.0239 | 0.1417 | 0.0467 | 0.0655 | 0.0913 | **0.0987** |
| TANGEM-Plain | 0.0155 | 0.0030 | 0.1201 | 0.0104 | 0.1518 | 0.0601 | 0.0022 | 0.0721 | **0.0214** | 0.5993 |
| TANGEM | **0.0038** | **0.0004** | **0.0548** | **0.0001** | **0.0155** | 0.0449 | **0.0007** | **0.0464** | 0.0397 | 0.7055 |

| | Pems04 | | | | | CiteSeer | | | | |
|---|---|---|---|---|---|---|---|---|---|---|
| | Degree | Clustering | Spectral | Orbit | Motif | Degree | Clustering | Spectral | Orbit | Motif |
| E-R Generator | 0.1324 | 1.8028 | 0.1321 | 0.2152 | 1.2456 | 0.0805 | 1.9977 | 0.0705 | 1.9939 | 1.9534 |
| B-A Generator | 0.3191 | 1.8998 | 0.3617 | 1.1361 | 1.4508 | 0.0823 | 1.9881 | 0.0439 | 1.2782 | 1.3310 |
| GraphVAE | 0.2260 | 1.2625 | 0.1827 | 1.3945 | 1.2705 | - | - | OOM | - | - |
| NetGAN | 0.1213 | 1.3081 | 0.1656 | 0.0226 | 0.3007 | 0.0051 | **0.1371** | 0.0245 | 1.3514 | 1.1846 |
| GraphRNN | 0.0417 | 0.4823 | 0.1522 | 0.0036 | 0.0259 | - | - | OOM | - | - |
| GRAN | 0.0110 | 0.7434 | 0.1191 | 0.0133 | 0.1415 | 0.0131 | 1.7133 | 0.0245 | 1.3703 | 1.3763 |
| DiGress | 0.1383 | 1.0884 | 0.1570 | 0.4537 | 1.4593 | - | - | OOM | - | - |
| TANGEM-Plain | 0.0407 | 0.3515 | 0.1410 | 0.0650 | 0.7994 | **0.0008** | 0.1515 | 0.0085 | 1.6221 | 1.6530 |
| TANGEM | **0.0012** | **0.2694** | **0.0637** | **0.0004** | **0.0048** | 0.0024 | 0.6963 | **0.0079** | **0.0799** | **0.0112** |

a fixed probability, and the Barabási–Albert (B–A) model (Barabási et al., 2002), which reproduces scale-free networks via preferential attachment. For deep learning baselines we included GraphVAE (Simonovsky & Komodakis, 2018), a variational autoencoder for latent-variable graph reconstruction; GraphRNN (You et al., 2018b), which sequentially generates nodes and edges with an RNN; GRAN Liao et al. (2019), which combines block-wise autoregression and self-attention; NetGAN Bojchevski et al. (2018), which uses random walks and adversarial training to learn graph distributions; and DiGress Vignac et al. (2023), a diffusion-based model that denoises to produce high-quality, diverse graphs. NetGAN is the only baseline able to learn from a single input graph, like our approach, whereas the others require multiple graphs. To ensure compatibility of our single-graph datasets (IBB1, IBB2, and Pems04) with these methods, we created augmented versions through random edge additions and removals.

## 4.3 EVALUATION METRICS

Following prior work You et al. (2018b); Liao et al. (2019), we assess TANGEM by comparing generated and original graphs using Maximum Mean Discrepancy (MMD) Gretton et al. (2012) over key structural properties: degree distribution, clustering coefficient, spectral characteristics, motif count, and orbit count. These metrics respectively capture connection frequencies, community structure, global connectivity via Laplacian eigenvalues, prevalence of recurring subgraphs, and roles of nodes within motifs. For each property, we compute MMD with a Gaussian kernel to measure the divergence between the distributions of the original and generated graphs, ensuring a robust evaluation of both local and global structural features. We additionally conduct visual inspection to qualitatively assess structural resemblance (see Appendix).

## 4.4 EVALUATING THE GENERATED GRAPHS

Table 1 presents a comparison of different graph generation models, including two variants of TANGEM. TANGEM-Plain represents a simpler version of TANGEM, where the transformer is trained on uniform random walks instead of temporal-aware biased walks. Still, TANGEM-Plain remains competitive with the baselines. TANGEM, on the other hand, shows strong performance against all baselines across every structural metric. On the IBB1 and Pems04 datasets, which are path-like, TANGEM surpasses all other methods in almost all metrics. This suggests that TANGEM is particularly effective at modeling path-like graphs (see also Fig. 3). However, some caution is

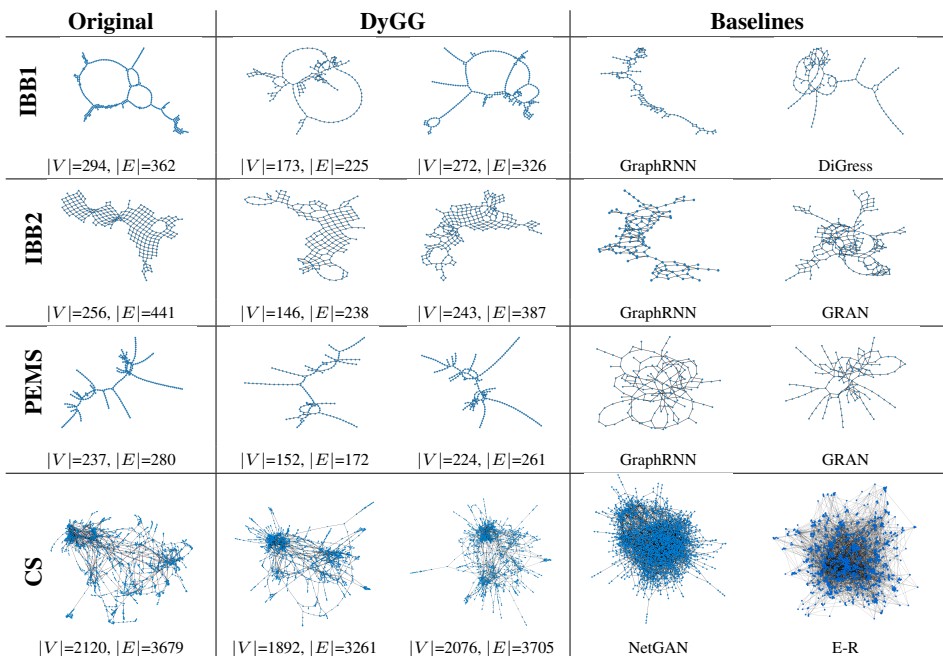

Figure 3: Comparison between original and generated graphs with different sizes. The first column shows the original graphs, while subsequent columns show generated graphs at varying sizes. The subfigures demonstrate that the generated graphs preserve similar structural characteristics to the originals despite differences in size. (More visual examples can be found in A.7)

needed, since such paths could be memorized by the model due to the explorative nature of the walks, which can nearly cover long paths. To investigate this, we generated graphs of different sizes and visualized them, as shown in Fig. 3. For the IBB2 dataset, no method clearly outperforms the others, though TANGEM remains competitive. On CiteSeer, NetGAN is the only comparable deep generative model, and TANGEM surpasses it on all metrics except clustering.

Following previous works (You et al., 2018b; Liao et al., 2019; Vignac et al., 2023), we further evaluate the performance of TANGEM via visual inspection. Fig. 3 visualizes the graphs generated by TANGEM and baselines, highlighting that TANGEM can capture and synthesize the real-world graphs with characteristics like path, grid, and community. For the CiteSeer graph, it can be clearly seen that the community structures and sparsity are well-preserved. Notably, TANGEM generates relatively large graphs within a highly acceptable training and inference time, whereas all deep graph generators but NetGAN fail to scale. As seen in the figure, although the NetGAN-generated CiteSeer graph weakly reflects community structure, it fails to represent the overall characteristics of the original graph.

To assess different walks we experimented using different types of walk sampling strategies and shared results in Fig. 4: (1) uniform random walk, which takes the next step randomly; (2) temporal-aware random walk, which gives weights to candidate nodes based on temporal correlation; (3) biased random walk ($q \ll 1$), which only prioritize exploration; (4) temporal-aware biased walk, which prioritizing exploration while favoring nodes that are temporally correlated. For the IBB1 and Pems04 graphs, incorporating explorative behavior (RW+ and BRW+) leads to better performance compared to their non-explorative counterparts (RW and BRW). Moreover, in both random walk and biased random walk settings, adding temporal awareness further improves performance. For the IBB2 graph, explorative behavior remains beneficial, as BRW and BRW+ outperform RW and RW+. However, temporal awareness does not provide any improvement and may even slightly degrade performance. A possible explanation is that the regular grid structure of IBB2 does not align with temporal correlations, making the usage of temporal features misleading. Finally, for the CiteSeer graph, most metrics benefit from both explorative behavior and temporal awareness, except

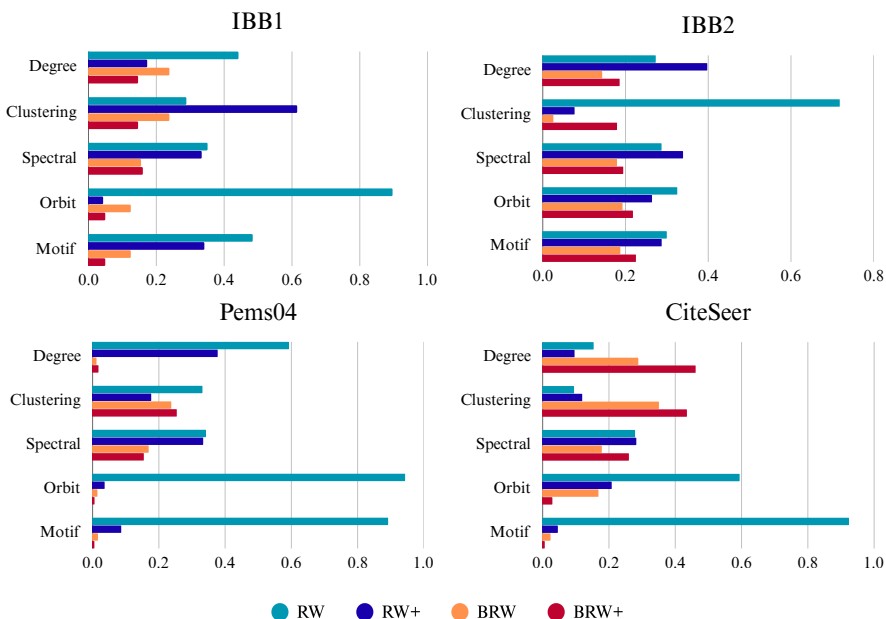

Figure 4: Comparison of random walk (RW), temporal aware random walk (RW+), biased random walk (BRW), and temporal aware biased random walk (BRW+). For better visualization, the values are normalized (see Appendix A.8 for the original values).

degree and clustering metrics. In particular, orbit and motif metrics show substantial improvement when both temporal-awareness and explorative behavior are incorporated.

**Limitations.** While TANGEM introduces a new framework for generating temporally attributed graphs with a fixed topology, it currently relies on several simplifying assumptions. First, it assumes that the set of nodes and the underlying edge skeleton are known in advance and remain stable across snapshots. This choice is well-suited to domains such as transportation or sensor networks, but limits direct applicability to settings where nodes and edges emerge frequently, such as social networks. As discussed above, this was a pragmatic choice to sharpen our focus and improve efficiency. An extension toward dynamic graphs, where edges or nodes may appear or disappear over time, is to train on the union of all edges observed across time and to learn a time-dependent edge-presence function. For example, during generation, walks could still be sampled on this union graph, while a neural head $g(u, v, t)$ would output a time-dependent edge-survival probability (e.g., Bernoulli) conditioned on temporal similarity $\rho(u, v)$, recency, and walk context, retrofitting edge addition and deletion without redesigning the core architecture. Second, TANGEM currently leverages a temporal similarity signal to guide walk sampling and message passing. Although this bias improves structural fidelity in homophilic or smoothly evolving signals, it may be less effective in non-homophilic graphs where connected nodes display dissimilar temporal patterns. Addressing these constraints opens several research avenues.

## 5 CONCLUSION

We presented TANGEM, a generator for temporally attributed graphs with a fixed topology. By injecting a learned temporal similarity into a transformer-based biased random walk, TANGEM directly couples signals and structure so that recurring co-activation patterns guide which motifs and spectral modes the generated graphs emphasize. Across benchmarks, it consistently improves structural metrics over strong static baselines. To keep this study focused and provide clear ablations, we have intentionally left extensions to dynamic graphs and non-homophilic settings for future work. Nonetheless, we believe TANGEM establishes a solid foundation for temporally attributed graph generation in fixed-topology regimes and serves as a stepping stone toward future models that also incorporate evolving nodes and edges.

REPRODUCIBILITY STATEMENT

The main paper specifies the problem setting and evaluation separation (Sec. 3: the definition of the temporally-attributed graph and fixed-topology regime; Sec. 4: structural vs. downstream metrics) and defines all metrics (clustering, spectral MMD) and baselines. The Appendix details the topology generator (temporal-similarity construction, transformer walk model, and hyperparameters), ablations, and the routine feature-sampling setup used only for analysis. Our anonymous code repository (link in Supplementary) includes: implementation of TANGEM's topology module, exact training/evaluation scripts for each dataset, configuration files with all hyperparameters, random seeds, and run commands; data preprocessing pipelines for time-series benchmarks (including splits and normalization); and scripts to compute all reported metrics and figures. We also release preprocessed datasets, instructions for environment setup (including requirements and container specifications), and notes on hardware/runtime in the public repository.

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

# A  APPENDIX

## A.1  IMPLEMENTATION DETAILS OF TANGEM

The TANGEM employs a transformer that processes sampled walks, treating graphs as sequences of tokens. The model uses learnable token and positional embeddings of size 64, with a context window defined by the block size. The architecture consists of 4 Transformer layers, each with 4 self-attention heads and a feed-forward network, connected through residual links and layer normalization. A dropout rate of 0.5 is applied. The final linear layer projects hidden states to the vocabulary space, and the model is trained with cross-entropy loss. During inference, the model generates graph sequences autoregressively by sampling from the predicted token distribution.

The AdamW optimizer is used for minibatch training, and each minibatch contains 128 tokenized walks. Depending on the input graph, walk length may differ (usually around 30-50). The default learning rate is set as 0.001, and the maximum number of epochs is set as 10,000; early stopping is applied.

## A.2  RUNNING TIME OF TANGEM

Training is conducted on a Google Colab L4 accelerator. The training time for CiteSeer is around 10 to 20 minutes, and the inference time for generating one CiteSeer graph is around 2 minutes.

During the generation of small graphs $|V| < 1000$, generation can be achieved in one long sequence. However, for larger graphs, multiple sequences can be sampled independently and then composed into a single graph. For CiteSeer ($|V| = 2120$) generation, we composed 3 independent sequences, each initialized from a randomly selected token.

## A.3  EXPERIMENTAL SETUP

We evaluated deep graph generation baselines using their official implementations and default settings: GraphRNN, GRAN, NetGAN, and DiGress. Since NetGAN relies on an older version of TensorFlow (1.x), we ran it locally, while the other models were executed in Google Colab notebooks for convenience. To provide training inputs for GraphRNN, GRAN, and DiGress, which require multiple graphs, we applied simple data augmentation (e.g., edge dropping) to generate additional graphs from the original one. For evaluation, each method generated 10 graphs, and we measured their similarity to the original graph using the MMD distance. The same evaluation procedure was applied consistently across all baselines.

## A.4  NOTATION TABLE

## A.5  SIMILARITY MATRIX

We utilize a pairwise similarity matrix $S \in \mathbb{R}^{|V| \times |V|}$ to integrate temporal node features, capturing their relationships and integrating this information into the walk sampling process as shown in Fig. 5. This allows the walks to reflect the more localized behavior of the nodes while maintaining exploration across the graph.

## A.6  HEAT DIFFUSION ON GRAPHS

In order to evaluate the performance of our temporal-aware biased random walk method on synthetic and benchmark static graphs (i.e., community and Citeseer), we generated synthetic node features from the heat diffusion process. An example of generated temporal node features has shown in Fig. 6.

The graph Laplacian matrix is used in modeling the heat diffusion process throughout a graph. The fundamental equation governing heat diffusion is given by

$$\frac{d\mathbf{h}}{dt} = -L\mathbf{h} \tag{9}$$

Table 2: Notation summary.

| Symbol | Description |
|---|---|
| $G = (V, E)$ | Underlying graph topology with node set $V$ and edge set $E$. |
| $X \in \mathbb{R}^{|V| \times T \times F}$ | Tensor of temporal node features; $T$ time steps, $F$ features per node per time step. |
| $x_i \in \mathbb{R}^{T \times F}$ | Temporal feature vector of node $i$. |
| $S \in \mathbb{R}^{|V| \times |V|}$ | Pairwise temporal similarity matrix computed from node features. |
| $S_{ij} = f(x_i, x_j)$ | Similarity between node $i$ and node $j$ defined by metric $f$. |
| $f(\cdot, \cdot)$ | Similarity metric (e.g., cosine similarity, correlation, RBF kernel). |
| $\rho(u, v) = S_{uv}$ | Temporal bias lookup function for nodes $u$ and $v$; extended in Eq. (7) to include second-hop neighbors. |
| $A$ | Adjacency matrix of the graph. |
| $L = D - A$ | Unnormalized Laplacian matrix; $D$ is diagonal degree matrix. |
| $p$ | Return parameter of Node2Vec-style second-order random walk. |
| $q$ | In–out (exploration) parameter of Node2Vec-style second-order random walk. |
| $\delta_{kv}$ | Shortest-path distance between nodes $k$ and $v$ (0, 1, or 2 for direct neighbors). |
| $\alpha_{pq}(k, v)$ | Search bias (Node2Vec) for moving from node $u$ to $v$ after visiting $k$. |
| $\alpha_q(k, v, u)$ | Modified search bias with temporal bias $\rho(u, v)$ (Eq. 5). |
| $s = \{v_1, \ldots, v_M\}$ | A node sequence (walk) of length $M$. |
| $P(s)$ | Joint probability of generating sequence $s$ (Eq. 8). |
| $M$ | Sequence length / maximum number of tokens to generate. |
| $N(v)$ | Set of neighbors of node $v$. |
| $\lambda$ | Hyperparameter controlling influence of second-hop neighbors in $\rho(u, v)$. |

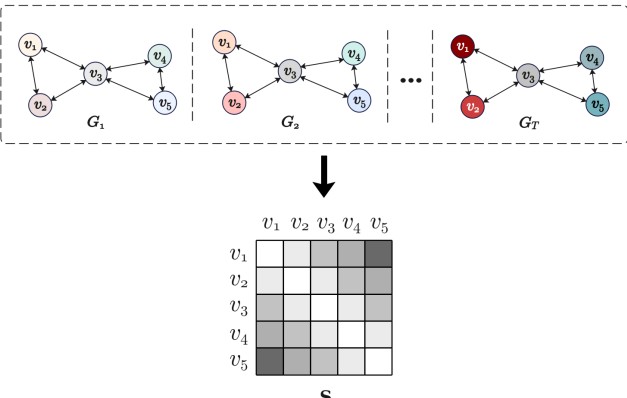

Figure 5: Similarity Matrix (S): similarity matrix using temporal node features across graph snapshots. Each entry reflects the similarity between nodes based on their temporal feature evolution over time.

where $h$ is a vector representing the heat value at each node, and $t$ represents the time. The unnormalized Laplacian matrix L is defined as

$$L = D - A \qquad (10)$$

where $A$ is the adjacency matrix of the graph and $D$ is diagonal degree matrix defined as

$$D_{ii} = \sum_j A_{ij} \qquad (11)$$

The solution of Eq. 9 is given by

$$\mathbf{h}(t) = e^{-tL}\mathbf{h}(0) \qquad (12)$$

where $e^{-tL}$ is knows as the heat kernel, and the $h_0$ is the vector of initial heat values associated with nodes.

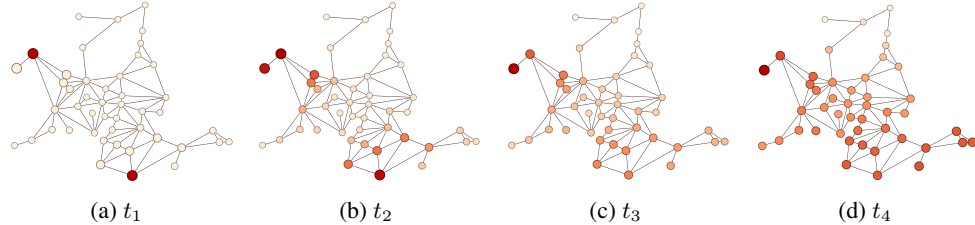

(a) $t_1$     (b) $t_2$     (c) $t_3$     (d) $t_4$

Figure 6: Example of heat diffusion on graphs: Red color represents high temperature at nodes.

Table 3: Original graph names are shown on the left. Each row has the original graph and four generated graphs by TANGEM.

## A.7 MORE VISUAL EXAMPLES BY TANGEM

We provide additional visual examples generated by TANGEM, showcasing its ability to capture complex patterns and temporal dynamics in graph structures, as seen in Table 3

## A.8 ASSESSING DIFFERENT WALK SAMPLING STRATEGIES

We compare four walkers: uniform Random Walk (RW), temporal-aware Random Walk (RW+), Biased Random Walk (BRW, Node2Vec-style with $q \ll 1$), and Temporal-Aware Biased Random Walk (BRW+, combining BRW and temporal similarity), and values are given in Table 4. Lower values indicate better alignment to the target graph for each structural signature (Degree, Clustering, Spectral, Orbit, Motif). Bold marks the best per column. All walkers share identical decoder, walk lengths, and stitching procedures; only the transition kernel changes (uniform vs. biased vs. temporal-weighted). We report means over fixed seeds; code, seeds, and per-run logs are provided in the Supplement. The following points provide additional observations and takeaways that complement the main text.

- Exploration helps almost universally. Comparing BRW to RW improves 19/20 cells across datasets (the only exception is Clustering on CiteSeer). A low $q$ discourages backtracking and covers the graph more evenly, improving degree/spectral/orbit/motif statistics that rely on broader coverage.
- Temporal awareness is beneficial but dataset-dependent. Relative to BRW, BRW+ wins 9/20 cells:

- **IBB1 (community structure):** BRW+ improves 4/5 metrics; temporal similarity re-inforces homophily, aiding Clustering, Orbit, and Motif. A slight drop on Spectral suggests over-emphasis on tightly similar regions can shift global eigenvalue mass.
- **IBB2 (grid-like regularity):** BRW outperforms BRW+ on all metrics; a regular lattice has weak assortative signal, so temporal bias misaligns with the underlying regular structure, hurting degree/clustering/spectral fidelity.
- **Pems04 (traffic sensors):** BRW+ excels on Spectral, Orbit, and Motif, while BRW keeps a tiny edge on Degree. Here temporal co-activation is meaningful, so injecting similarity improves higher-order patterns.
- **CiteSeer (citation network):** BRW+ improves Orbit and Motif (role/motif statistics), but BRW remains better on Degree and Spectral, and uniform RW best preserves Clustering. Exploration can reduce triangle closure locally, explaining the clustering dip.

- Adopting an explorative bias(low $q$) as the default is sensible, as it consistently improves structural fidelity.

- Applying temporal bias is beneficial when temporal signals align with structural homophily or co-activation (e.g., traffic or community graphs), but it should be reduced or avoided on highly regular or weakly assortative graphs (e.g., grids).

- Different signatures react differently. Clustering looks at how many triangles (three nodes all connected) are formed locally. This favors simple random walks that stay close to the starting point, because they are more likely to see and preserve those local triangles. Orbit, Motif, and Spectral metrics look at larger patterns and the graph's overall shape. These benefit from walks that explore more widely (biased walks with low $q$) and, when it makes sense, use temporal information to steer the walk toward nodes that behave similarly over time.

Table 4: Comparison of random walk (RW), temporal aware random walk (RW+), biased random walk (BRW), and temporal aware biased random walk (BRW+).

| | IBB1 | | | | | IBB2 | | | | |
|---|---|---|---|---|---|---|---|---|---|---|
| | Degree | Clustering | Spectral | Orbit | Motif | Degree | Clustering | Spectral | Orbit | Motif |
| Random Walk (RW) | 0.0115 | 0.0030 | 0.1201 | 0.0104 | 0.1518 | 0.0661 | 0.0028 | 0.0685 | 0.0592 | 0.9392 |
| T-A Random Walk (RW+) | 0.0045 | 0.0064 | 0.1142 | 0.0005 | 0.1071 | 0.0961 | 0.0003 | 0.0809 | 0.0481 | 0.9003 |
| Biased Random Walk (BRW) | 0.0062 | 0.0006 | **0.0530** | 0.0006 | 0.0391 | 0.0347 | 0.0001 | 0.0428 | 0.0351 | 0.5875 |
| T-A Biased Random Walk (BRW+) | **0.0038** | 0.0004 | 0.0548 | 0.0001 | 0.0155 | 0.0449 | 0.0007 | 0.0464 | 0.0397 | 0.7055 |

| | Pems04 | | | | | CiteSeer | | | | |
|---|---|---|---|---|---|---|---|---|---|---|
| | Degree | Clustering | Spectral | Orbit | Motif | Degree | Clustering | Spectral | Orbit | Motif |
| Random Walk (RW) | 0.0402 | 0.3515 | 0.1411 | 0.0653 | 0.7994 | 0.0008 | 0.1515 | 0.0085 | 1.6221 | 1.6534 |
| T-A Random Walk (RW+) | 0.0256 | **0.1875** | 0.1375 | 0.0025 | 0.0774 | 0.0005 | 0.1909 | 0.0076 | 0.5686 | 0.0819 |
| Biased Random Walk (BRW) | **0.0008** | 0.2517 | 0.0697 | 0.0012 | 0.0145 | 0.0015 | 0.5612 | **0.0048** | 0.4595 | 0.0426 |
| T-A Biased Random Walk (BRW+) | 0.0012 | 0.2694 | **0.0637** | **0.0004** | **0.0048** | 0.0024 | 0.6963 | 0.0079 | **0.0799** | **0.0112** |

## A.9 EVALUATING DOWNSTREAM NODE SAMPLING ON GENERATED TOPOLOGIES

Additionally, we used generated Pems04 graphs from TANGEM and GRAN (the one of the strongest baseline), and mapped the traffic features from the original dataset. Then, we trained an attention-based spatial-temporal graph convolutional network Guo et al. (2019) to learn the traffic patterns and predict 5 days of traffic data. The results for TANGEM and GRAN are visualized in Fig. 7a and Fig. 7b. This ablation study aims to test whether a generated topology supports realistic temporal behavior when real signals are placed on it. We took graphs generated by TANGEM and by GRAN; then, we mapped the original PEMS04 traffic signals onto each generated graph; then, we trained the same Attention-Based Spatial-Temporal GCN Guo et al. (2019) to forecast 5 days of traffic. PEMS is a fixed-topology domain, directly take from real road network. If the generated structure reflects real temporal co-activation (who tends to rise/fall together), a standard predictor should forecast more accurately on that structure. Here, in 7a, forecasts on TANGEM graphs track the ground truth more closely than forecasts on GRAN graphs, as seen in Fig. 7b. Because the predictor and signals are held constant across methods, the performance gap is attributable to the generated topology, i.e., the structure TANGEM produces provides a better substrate for temporal propagation

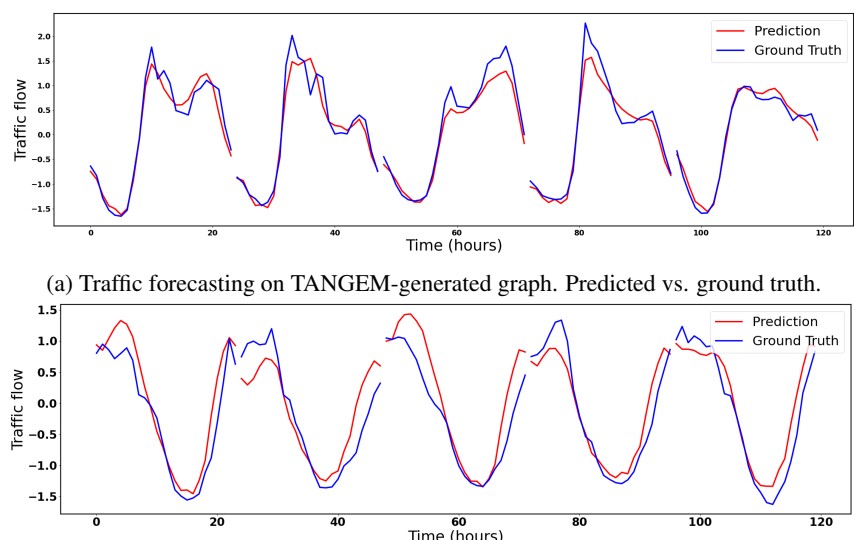

(a) Traffic forecasting on TANGEM-generated graph. Predicted vs. ground truth.

(b) Traffic forecasting on GRAN-generated graph. Predicted vs. ground truth.

and aggregation. This experiment shows that such signal-guided structure is not just cosmetically closer in static metrics; it also improves downstream temporal consistency.

