# OpenReview forum: "Graph Generation via Temporal-Aware Biased Walks"
_ICLR.cc/2026/Conference — ICLR 2026 Conference Withdrawn Submission_

### Official Review · Reviewer_Kcn5 · 2025-10-30

**Soundness:** 2
**Presentation:** 3
**Contribution:** 2
**Rating:** 2
**Confidence:** 4

**Summary:**

This paper proposes TANGEM, a method for graph generation that integrates temporal similarity into
biased random walks. The resulting node sequences are modeled autoregressively with a Transformer decoder,
from which graphs are generated. Experiments are conducted on a few small temporal datasets and compared
with several classical graph generation methods.

**Strengths:**

This manuscript is well organized, presenting a logically coherent structure.
The idea of coupling node signal similarity and random walk–based graph generation is interesting and intuitive.

**Weaknesses:**

1. The proposed method is essentially Node2Vec with a temporal similarity bias and a Transformer decoder. It does not model dynamic topology, temporal dependencies, or event sequences as in true temporal graph learning works (e.g., TGAT, DyGFormer). The claimed contribution of “temporal-aware generation” is conceptually shallow and incremental.
2. Outdated and incomplete baselines. No comparison is made with modern temporal graph generators. Without stronger baselines, the experimental significance is limited.
3. Figures are of low quality, and many contain LaTeX errors (e.g., stray $$ in Fig. 1).
4. Experiments too small-scale and weakly analyzed. Datasets are small (hundreds of nodes). There is no scalability or runtime analysis, and training is done on Colab for minutes, suggesting a toy-level setup. No ablation explaining why temporal bias helps or when it might fail.
[1] Yu L, Sun L, Du B, et al. Towards better dynamic graph learning: New architecture and unified library[J]. Advances in Neural Information Processing Systems, 2023, 36: 67686-67700.
[2] Xu D, Ruan C, Korpeoglu E, et al. Inductive representation learning on temporal graphs[J]. arXiv preprint arXiv:2002.07962, 2020

**Questions:**

1. Does the model update node embeddings over time, or is the time dimension only used for biasing random walks?
2. Could the authors clarify whether the approach can handle temporal edge additions/removals or only static graphs with time features?
3. How does this method differ conceptually from applying Node2Vec on a static graph with time as a node feature?
4. Without dynamic updates, why is this considered a temporal graph learning method rather than static graph generation?

---

### Official Review · Reviewer_pGCm · 2025-10-30

**Soundness:** 2
**Presentation:** 2
**Contribution:** 2
**Rating:** 2
**Confidence:** 4

**Summary:**

The paper proposes a method for graph generation. First of all, random walks are extracted, where the selection of the next node in a walk is guided by the temporal features of the nodes, specifically transitions are more likely between nodes with similar temporal behavior. These temporally biased random walks are then used to train a Transformer model, which is later employed to generate new random walks. The generated walks are finally combined to form a synthetic graph. The approach couples temporal dynamics with structural properties, and experiments on real and synthetic datasets show improvements in structural metrics (degree, clustering, spectral, motif MMD) over baselines such as GraphRNN, NetGAN, and DiGress.

**Strengths:**

The topic of graph generation is important and relevant for many applications. It is interesting that the paper proposes to use the information contained in temporal node features to guide the generation of the graph topology.

**Weaknesses:**

**Ambiguity of the task and unclear motivation**
The main concern lies in the aim of the paper. The task is defined as generating graphs from a graph with a fixed topology and temporally varying node features (the authors call such graphs “temporally attributed graphs,” but this concept already exists in the literature and is commonly referred to as spatiotemporal graphs [1]). However, the proposed model ultimately generates only a static graph structure—temporal features are not generated or represented in the output. As a result, the problem reduces to generating a static graph from a spatiotemporal one.

In general, generating synthetic graphs similar to observed ones can be meaningful for data augmentation, privacy preservation, or simulation purposes, when the goal is to sample from the same underlying distribution of graphs. But in this case, what is the motivation for generating a single static synthetic graph from one spatiotemporal graph? What is the expected use case of such a generated graph? Since temporal features are discarded in the output, it is unclear what practical or scientific insight is gained. The authors should provide a clear rationale or concrete examples of scenarios where this task is useful.

**Confusing exposition**
 The methodological explanation is confusing and makes it difficult to follow an otherwise conceptually simple idea. In particular, a) the formal mathematical definition of the task to be solved is missing; b) random walks and the temporal bias used are introduced before explaining their role in the pipeline, leaving the reader disoriented; 3c the transformer component is described without clearly specifying what it takes as input and how its outputs are transformed into a new graph. Overall, the idea itself is not complex, but the writing makes it hard to understand.

**Similarity function**
Equation (1) defines the temporal similarity as $S_{ij} = f(x_i, x_j)$, but in the main text it is never stated what this function $f$ actually is. In the appendix, you mention that it could be a cosine similarity or a correlation function, but this is not clearly justified or discussed. It is important to remember that $x_i$ and $x_j$ are not simple vectors --- they are time series, carrying temporal structure and dependencies. Measuring similarity between two time series is a non-trivial task, and reducing it to a simple vector similarity (like cosine) is quite limiting. What function $f$ did you actually use in your experiments? Did you consider more appropriate similarity measures that account for the temporal nature of these features (e.g., DTW, temporal kernels, or cross-correlation over time)?  The same applies to the parameter $q$ in the random walk: its value is never specified, nor is there any sensitivity analysis. How was $q$ chosen, and have you tested how sensitive the model is to its variation?

**Choice of datasets**
Although several standard spatiotemporal graph datasets exist (e.g., METR-LA [2], CER-E [3], AQI [4], ENG-RAD [5]), the paper instead uses a static benchmarks such as CiteSeer, where synthetic temporal features are artificially generated via a heat diffusion process. This weakens the empirical validation, since it does not demonstrate the method’s utility in realistic spatiotemporal scenarios. The authors should justify this choice.

**Minor issues**
In the abstract, the phrase “we introduce temporally attributed graphs, namely TANGEM” is inaccurate as the paper introduces a method, not a new type of graph. Figure 1 is never explicitly referenced in the text and still contains LaTeX artifacts. Appendix A.4 is empty (it seems it should contain the table on the following page, but without text referring to it, it appears blank).


[1] Cini, Andrea, et al. "Graph deep learning for time series forecasting." ACM Computing Surveys 57.12 (2025): 1-34.
[2] Zeng Li, Clifford Lam, Jianfeng Yao, and Qiwei Yao. 2019. On testing for high-dimensional white noise. The Annals of Statistics 47, 6 (2019), 3382–3412.
[3] Commission for Energy Regulation. 2016. CER smart metering project - electricity customer behaviour trial, 2009- 2010 [dataset]. Irish Social Science Data Archive. SN: 0012-00 (2016). Retrieved from https://www.ucd.ie/issda/data/ commissionforenergyregulationcer
[4] Yu Zheng, Xiuwen Yi, Ming Li, Ruiyuan Li, Zhangqing Shan, Eric Chang, and Tianrui Li. 2015. Forecasting finegrained air quality based on big data. In Proceedings of the 21th ACM SIGKDD International Conference on Knowledge Discovery and Data Mining. 2267–2276.
[5] Ivan Marisca, Cesare Alippi, and Filippo Maria Bianchi. 2024. Graph-based forecasting with missing data through spatiotemporal downsampling. In Proceedings of the International Conference on Machine Learning. PMLR, 34846– 34865.

**Questions:**

Some questions (already presented above):

- What is the motivation for generating a single static synthetic graph from one spatiotemporal graph? What is the expected use case of such a generated graph?

- What function $f$ did you actually use in your experiments? Did you consider more appropriate similarity measures that account for the temporal nature of these features (e.g., DTW, temporal kernels, or cross-correlation over time)?

- How was $q$ chosen, and have you tested how sensitive the model is to its variation?

---

### Official Review · Reviewer_KtXi · 2025-10-31

**Soundness:** 3
**Presentation:** 3
**Contribution:** 2
**Rating:** 4
**Confidence:** 3

**Summary:**

This paper introduces TANGEM, a temporally-attributed graph generation method based on biased temporal random walks. The authors assume that the temporal correlation of features is stronger between structurally closer nodes and thus compute node similarity to serve as an inductive bias when assigning walk probabilities. Then, the authors use an LLM-like autoaggressive framework to generate node sequences, which subsequently compose the graph structures. Extensive experiments demonstrate the effectiveness of the proposed method.

**Strengths:**

1. The idea of integrating the graph homophily into the random walk process is interesting.
2. The proposed method exhibits expressive results compared to the baselines, as well as the case study.
2. This paper is well-written and easy to follow.

**Weaknesses:**

1. The novelty of the proposed method seems to be limited, which lacks technical contribution within the community. The idea of similarity-based random walk has been largely studied by existing works [1, 2]. Moreover, the proposed approach seems loosely related to temporally-attributed graphs. In such graphs, node features evolve dynamically over time, but the proposed method merely exploits current node similarity without incorporating the long-term temporal evolution of node attributes into the sampling process.
2. The paper lacks a clear and rigorous task definition. What exactly distinguishes graph generation on temporally-attributed graphs from graph generation on static or other types of graphs? Are there unique challenges inherent to temporally-attributed graphs that motivate the proposed approach? Why are existing graph generation methods not directly applicable or effective in this domain?
3. The hyperparameter, such as λ or M, should be analyzed to understand the sensitivity of the proposed method.
4. The baselines used for comparison appear to be outdated. I encourage the authors to include more recent and competitive baselines for comparison.
5. How does the proposed method perform when the assumption of graph homophily fails? Does it still work well?

[1] SARW: Similarity-Aware Random Walk for GCN
[2] Similarity and Self-similarity in Random Aalk with Fixed, Random and Shrinking steps

**Questions:**

See weaknesses

---

### Official Review · Reviewer_ExbD · 2025-11-01

**Soundness:** 1
**Presentation:** 2
**Contribution:** 1
**Rating:** 2
**Confidence:** 4

**Summary:**

The authors of this work propose a random walk-based model to generate graphs based on so-called temporally attributed graphs, i.e. graphs with a static topology and node attributes that evolve in discrete time. For this, they propose a biased second-order random walk on the static graph topology, which integrates two mechanisms to bias transition probabilities. The first strategy uses topological features inspired by node2vec, where a bias parameter of second-order transition probabilities influences the exploration/exploitation behavior of the walk based on the static topology of the graph. The second mechanism is based on a temporal similarity matrix that captures the pairwise alignment of node attributes in time serie data capturing evolving node attributes in a static graph. The entries of this temporal correlation matrix are used as a "temporal bias" p(u,v) that influences the probability of a random walker currently residing in node u to move to a neighbor node v.

The actual graph generation is then considered as a autoregressive generation of a node sequence, where a transformer model is trained on node sequences sampled from the biased random walk model. As far as I understood this part of the work (see my question on ths below) the generated sequences are then aggregated into a static graph, thus discarding any sequential patterns in the biased random walk sequences that may have been captured by the second-order random walk model or the transformer model.

The authors evaluate their approach in four rather small data sets, three empirical graphs on traffic time series in road networks (256, 294 and 237 nodes) and a third one using a simple diffusion model to generate temporal signals in a static citation network (2120 nodes). The results of the graph generation are compared to simple random graph models and deep graph generation models, comparing degree distributions, clustering coefficient, spectral properties and motifs. The results show rather mild improvements in performance for the proposed model in most metrics for three of the data sets, where differences to a simpler model without temporal bias are particularly small.

**Strengths:**

[S1] The paper is generally well-written and includes an illustration that helps to graps the intution for the temporally biased random walk.

[S2] The proposed model is conceptually simple and the idea to use a matrix of node similarities extracted from evolving atttributes is neat. It seems rather obvious to use this to bias a random walk but has - to the best of my knowledge - not been addressed in this form.

**Weaknesses:**

[W1] The motivation for the proposed model, i.e. the precise research gap that the authors address, as well as the key contributions of the work should be made clearer in the introduction, see details in Q1.

[W2] The motivation for some model ingredients, and the underlying assumptions of the dynamical process, are not really laid out in the paper, which makes it difficult to get an intuition for the question to which types of systems or data the model can be reasonably applied. See my questions Q2 and Q3 below.

[W3] It is unclear why the transformer model (which captures sequential patterns in the generated walks) is used to generate node sequences, which are apparently simply aggregated into a static graph, see question Q4 and Q5 below.

[W4] The experimental evaluation is weak and inconclusive, considering the rather small data sets used, the definition of evaluation metrics, and the choice of baseline models. It is unclear what the key component - the temporal bias in the second-order random walk - contributes to the results, see my question Q6 - Q8 below.

**Questions:**

[Q1] Referring to W1 above, please clarify early on in the paper what is expected input and output of your generative model. I understood that the input is a time-attributed graph, while the output is a simple static, unweighted and directed graph (it is not even made clear whether this is weighted, unweighted, directed or undirected). Is this assumption correct? If my assumption is correct, then the statement in the conclusion that TANGEM is "a generator for temporally attributed graphs with fixed topology" is misleading, since it does not actually generate temporally attributed graphs (it rather uses such graphs to generate static graphs). Please clarify.

[Q2] I did not grasp the intuition behind Eq.7, which extends the temporal bias to the second-order neighborhood of nodes. I would argue that the influence of second-order influence between nodes should already be subsumed in the temporal similarity matrix, which is based on the temporal alignment of node attributes in the underlying graph - assuming that the influence between nodes transitively expands beyond the immediate neighbors. Could you explain why second-order influence is explicitly modeled and why we do not need to extend this to, e.g., third-order neighbors?

[Q3] Related to W2 above, I think the assumptions about the process underlying the node attribute dynamics captured in the temporal similarity matrix should be laid out more clearly. As an example, for the traffic data used in the experimental evaluation, there is an underlying process that moves a conserved quantity (vehicles) through a static graph and those vehicles are likely to move on trajectories that cannot be modelled by a first-order Markov model. For the CiteSeer data set, the authors used a particularly simple heat-diffusion process, which has the Markov property and follows a linear dynamics. The assumptions baked into those processes are never made clear, and clarifying for what processes the authors deem their model suitable would greatly help the reader to appreciate the contributions of this work. Could you clarify what you assume about the underlying process?

[Q4] Related to W3, while I liked the idea of the temporally biased random walk model, the authors lost me when they describe the transformer-based graph generation process. First of all, it is a major omission that the authors never clearly state what exact output their model generates. My assumption is that it is a simple, unweighted possibly directed static graph but this should be made clear. Second, if I my assumption is correct, I do not understand what the transformer model (and the biased second-order random walk) which are both modelling sequential patterns contribute, since the generated sequences are anyway simply aggregated into a graph, which discards any sequential patterns that involve more than two consecutive nodes. Could the authors explain this important aspect of their approach?

[Q5] The use of the transformer model is motivated from a computational efficiency point of view, however there is no evaluation of this. Especially for the small examples used in the evaluation, there will definitely not be any issues of sampling graphs from a biased second-order random walk (which can be done using the node2vec implementation for graph that are larger by many orders of magnitude). Could the authors clarify this?

[Q6] The evaluation metrics are unclear and make it virtually impossible to interpret the results. In section 4.3 the authors state that they calculate the maximum mean discrepancy (MMD) for several "structural properties". First, I would have preferred a clear motivation for the choice of MMD as a measure. Second, some of those properties are distributions (e.g. degree distribution) where a MMD makes sense, but other properties are single values (e.g. the clustering coefficient) where it is unclear what the value means. Moreover, even the stated intepretation of the metrics is wrong, sincde the authors state that the clustering coefficient captures "community structure", which is clearly not the case. Could it be that the authors have confused "cluster patterns" in the sense of community structure with "clustering" in the sense of closed triangles or transitivity in graphs? On another note, the authors should clearly define their metrics (or at least cite the respective works).

[Q7] I was very confused by Figure 3, where the textual description and figure caption seem to suggest that the figure contains graphs generated by the proposed TANGEM model, but where I cannot find this in the figure. Also, what does DyGG refer to? Is this a leftover from a previous version of the manuscript and it should read TANGEM?

[Q8] A key question that I had while reading the experimental evaluation is what the actual contribution of this work, the temporally biased second-order random walk, contributes to the model. This could be seen as an ablation study and as far as I understood the authors, this is somehow addressed in Figure 4, where different random walk strategies are compared. In particular, a comparison between the BRW and BRW+ seems interesting, as I understood that BRW+ essentially corresponds to the proposed model (this should be clearly stated) while BRW corresponds to the second-order random walk in node2vec with fixed parameter q and no temporal bias. First of all, it is not clear how this parameter q has been chosen for the experimental results. Second, the results seem to suggest that the differences between the biased random walk with and without temporal bias are rather small (it is also unclear whether they are even significant since the measures have been normalized, no variance is given and there is no statement on how many instances were compared). This potentially diminishes the key contribution of the work, and should thus be made clear.

---

### Note · Authors · 2025-11-21

I have read and agree with the venue's withdrawal policy on behalf of myself and my co-authors.